# Effect of High-Intensity Focused Electromagnetic Technology in the Treatment of Female Stress Urinary Incontinence

**DOI:** 10.3390/biomedicines12122883

**Published:** 2024-12-18

**Authors:** Cheng-Yu Long, Kun-Ling Lin, Jian-Lin Yeh, Chien-Wei Feng, Zi-Xi Loo

**Affiliations:** 1Department of Obstetrics and Gynecology, Kaohsiung Medical University Hospital, Kaohsiung Medical University, Kaohsiung 80708, Taiwan; 830263@kmuh.org.tw (C.-Y.L.); 960233@kmuh.org.tw (K.-L.L.); 1060477@kmuh.org.tw (J.-L.Y.); 1080532@kmuh.org.tw (C.-W.F.); 2Department of Obstetrics and Gynecology, Kaohsiung Municipal Siao-Gang Hospital, Kaohsiung Medical University, Kaohsiung 80708, Taiwan; 3Graduate Institute of Medicine, College of Medicine, Kaohsiung Medical University, Kaohsiung 80708, Taiwan; 4Center for Cancer Research, Kaohsiung Medical University, Kaohsiung 807378, Taiwan

**Keywords:** high-intensity focused electromagnetic technology, stress urinary incontinence, urodynamics, ultrasound topography

## Abstract

**Background**: The aim of the study was to assess the effect of high-intensity focused electromagnetic (HIFEM) technology in the treatment of female stress urinary incontinence (SUI). **Materials and Methods**: 20 women with SUI were delivered a treatment course with HIFEM technology. Patients attended 6 therapies scheduled twice a week. Validated questionnaires were assessed, including the overactive bladder symptoms score (OABSS), urogenital distress inventory-6 (UDI-6), incontinence impact questionnaire-7 (IIQ-7), international consultation on incontinence questionnaire (ICIQ), and valued living questionnaire (VLQ). Some urodynamic parameters, such as maximum flow rate (Qmax), residual urine (RU), and bladder volume at first sensation to void (Vfst). Bladder neck mobility in ultrasound topography was also collected pre- and post-treatment at 1- and 6-month follow-up visits. **Results**: HIFEM treatment significantly improved SUI symptoms on pad tests from 4.2 ± 5.5 to 0.6 ± 1.3 and patients’ self-assessment in the 6-month follow-up. Additionally, the data from urinary-related questionnaires, including OABSS (5.3 ± 3.9 to 3.9 ± 3.6), UDI-6 (35.7 ± 22.3 to 15.2 ± 10.6), IIQ-7 (33.1 ± 28.7 to 14.3 ± 17.2), and ICIQ (9.4 ± 5.0 to 5.4 ± 3.6), all showed a significant reduction. Then, the analysis of the urodynamic study revealed that only maximum urethral closure pressure (MUCP) (46.4 ± 25.2 to 58.1 ± 21.2) and urethral closure angle (UCA) (705.3 ± 302.3 to 990.0 ± 439.6) significantly increased after the six sessions of HIFEM treatment. The urethral and vaginal topography were performed and found that HIFEM mainly worked on pelvic floor muscles (PFM) and enhanced their function and integrity. **Conclusions**: The results suggest that HIFEM technology is an efficacious therapy for the treatment of SUI.

## 1. Introduction

Stress urinary incontinence (SUI) is defined as the involuntary leakage of urine with multiple causes, including physical effort or exertion, and approximately 30–40% of women suffer from SUI induced by pelvic organ prolapse (POP) [1,2]. It is still considered a relatively common condition with a prevalence ranging from 1.9 to 31.8% [3,4]. Although SUI is not a life-threatening disease, it also presents a significant social burden and leads to negative impacts on women’s quality of life [5]. Treatment of SUI is divided into surgery or nonsurgical options depending on the severity of symptoms. Nonsurgical methods included pelvic floor exercise, which could strengthen muscles surrounding the urethra or vagina, or a pessary [6,7].

The guidelines from the American Urological Association and Society of Urodynamics mentioned that the surgical treatment options of SUI listed included autologous fascia pubovaginal sling, Burch colposuspension, and mid-urethral slings [8]. However, the treatments mentioned above have their own complications. Some peri-operative complications were listed in Table 1. Additionally, long-term complications such as pain or dyspareunia may occur in approximately 25% of cases [9]. Thus, a novel treatment is urgently needed, especially in noninvasive procedures.

Some devices or experimental therapies were developed, such as pelvic floor muscle training, fractional CO_2_ laser therapy, high-intensity focused ultrasound, high-intensity focused electromagnetic (HIFEM), and platelet-rich plasma therapy to deal with SUI [14]. All of them worked through different mechanisms of action. Additionally, HIFEM focuses on recovering the function of pelvic floor muscles (PFM) [15]. PFM is a bunch of muscles that can support the organs in the pelvic floor and control continence. They could possibly lose function due to aging, menopause, pregnancy, childbirth, prostate cancer treatment, obesity, and the straining of chronic constipation [16].

The research on magnetic field therapy in human disease began with the study created by Bickford in 1965, and the application of magnetic stimulation on SUI was started by Galloway et al. in 1999 [17]. In this research, they first provided magnetic stimulation (MS) (5 Hz plus 50 Hz treatment) under a chair seat, and the treatments were performed twice a week for 6 weeks. The results showed that the pad test significantly reduced after the treatment in the 3-month follow-up, and leakage episodes also obviously decreased. The pad test changed from 2.5 to 1.3. Then, Fujishiro et al., 2000, used a 30-min stimulation repetition of 15 Hz in 5 s per minute MS once in SUI patients and examined the urodynamic change. The result showed that the maximum urethral closure pressure, frequency of UI, and quality of life were significantly improved in the one-week follow-up [18]. Moreover, Manganotti et al., 2007, also demonstrated a 15-min stimulation repetition cycle of 15 Hz in 3-s per minute MS six sessions in 2 weeks in SUI patients. The result depicted that quality of life (QoL), severity of SUI, and pad test changed in a 1-month follow-up [19]. However, the excitation location mentioned above focused on sacral roots instead of pelvic floor muscles. Furthermore, Gilling et al., 2009, used a 10-min stimulation at 10 Hz followed by a 10-min stimulation MS at 50 Hz accompanied with PFMT three times in 6 weeks in SUI patients. Their data showed that there was no more effective overall than sham treatment in this patient group in the 6-month follow-up [20].

HIFEM technology could stimulate PFM contractions in a higher frequency than normal Kegel exercises by focusing on the neuromuscular tissue. The electric currents from HIFEM could work on the whole pelvic floor. It would depolarize the motor neurons in PFM, and the constant stimulation of the muscle could possibly help patients regain their neuromuscular control [15,21]. These contractions surpass those achievable through voluntary muscle activation, promoting muscle strengthening, enhanced neuromuscular control, and restoration of pelvic floor function. The mechanism involves the depolarization of motor neurons, leading to repeated and controlled muscle contractions, which improve the structural support of the urethra and reduce urinary incontinence symptoms. The above studies mentioned that HIFEM technology might benefit SUI patients. However, the current studies focus on investigating the improvement of SUI-related symptoms of HIFEM. No study has been conducted to further urodynamics or urethral topography caused by HIFEM technology.

Therefore, our current study intended to comprehensively investigate the effect of HIFEM on SUI patients in four aspects: 1. Urinary-related symptoms included OABSS, UDI-6, IIQ-7, ICIQ-SF, VLQ; 2. urodynamic change included Qmax, RU, MCC, Pdet, MUCP, FUL; 3. urethral topography included bladder neck mobility, the urethral area in resting or strain condition with transvaginal ultrasound; 4. vaginal topography included vaginal width, vaginal area, and levator hiatus with transvaginal ultrasound. Our study contributes to translational medical research by evaluating a novel therapeutic strategy aimed at addressing the pathophysiology of SUI. Additionally, it provides mechanistic insights into the role of neuromodulation in pelvic floor muscle rehabilitation, supported by urodynamic and imaging-based analyses. Finally, the findings underscore the potential of HIFEM technology in regenerative medicine, offering a noninvasive alternative to existing treatments for SUI.

## 2. Materials and Methods

Twenty female participants were recruited in this study, and one of the participants lost F/U. Female patients were referred by primary care physicians if they met the inclusion criteria from Nov 2021 to March 2022 at the Department of Obstetrics and Gynecology, Kaohsiung Medical University Hospital. All patients were divided into three grades based on their clinical severity (ICIQ). The grading system was referenced from Hilton et al. (1991) [22] as follows: grade I (mild), urinary incontinence when coughing or sneezing; grade II (moderate), urinary incontinence when running or lifting objects off the floor; and grade III (severe), urinary incontinence when walking or climbing the stairs. The inclusion criteria included the women with bladder neck hypermobility (urethral hypermobility, type II SUI). The exclusion criteria included the followings: (1) urinary tract infection (2) comorbidities relevant to OAB (diabetes mellitus, spinal cord injury, stroke, or neurogenic diseases), (3) severe cardiovascular diseases, (4) coagulopathy, (5) liver failure, (6) renal failure, (7) chronic urinary inflammation (interstitial cystitis, urethral syndrome or painful bladder syndrome), drug or alcohol abuse in the past 12 months, (8) lower urinary tract surgeries in the past six months, (9) urinary catheterization, urologic malignancy, gross hematuria, significant bladder outlet obstruction, kidney stones, chronic pelvic pain, or inability to comprehend or comply with instructions [23]. We performed HIFEM treatment with the BTL EMSELLA (BTL Industries Inc., Boston, MA) device. BTL EMSELLA uses HIFEM technology for PFM strengthening and reduction of SUI. The device is composed of a power generator and a circular coil mounted in the seat of the chair. The HIFEM treatment is performed twice a week and sustained for three weeks. The information of the device is listed as below: https://btlaesthetics.com/en/for-providers/emsella-providers (accessed on 17 December 2024). Each treatment lasts for 28 min. The device uses a chair-style applicator in which the fully clothed patients can directly sit on the center of the stimulation coil, which generates a 2.5 Tesla magnetic field. The rationale behind the frequency choice in HIFEM technology is to optimize the induction of supramaximal muscle contractions and fat reduction. The frequency is selected to ensure effective penetration and interaction with PFM. This is achieved by using low-frequency electromagnetic waves that penetrate deep into the muscle tissue [24]. The frequency is calibrated to stimulate muscle fibers effectively, promoting increased muscle mass and strength, which is beneficial for both aesthetic and functional improvements [25].

The assessments of female lower urinary tract symptoms were listed as follows: overactive bladder symptom score (OABSS), urinary distress inventory-6 (UDI-6), incontinence impact questionnaire (IIQ-7), international consultation of incontinence questionnaire-short form (ICIQ-SF), and valued living questionnaire (VLQ). OABSS was referred to by Homma et al., 2006 [26]. The questionnaire concluded with four questions related to OAB (the maximum score range is from 2 to 5). The total score ranged from 0 to 15 points, and the higher points meant severe symptoms. UDI-6 and IIQ-7 were referred to from previous literature [27,28]. The ICIQ contained four questions (the maximum score range is from 5 to 10), and the total score is 21. The VLQ is an instrument that taps into 10 valued domains of living [29].

Urodynamic studies, including noninstrumented uroflowmetry, filling and voiding cystometry, and urethral pressure profilometry, were performed using a 6-channel urodynamic monitor (MMS; UD2000, Enschede, The Netherlands). Any uninhibited detrusor contraction during filling cystometry was deemed positive for detrusor overactivity.

Three-dimensional (3-D) transperineal ultrasound was performed for urethral and vaginal topography in women with SUI who received HIFEM treatment by well-trained physicians (Cheng-Yu Long, Kun-Ling Lin, or Zi-Xi Loo). The bladder neck mobility and urethral area were measured as in the previous study [30]. The Volusion General Electric Sonography 730 Expert (GE, Healthcare Ultrasound, Zipf, Austria) was used with a 3.5-MHz curved linear array. The transducer was placed between the labia majora and underneath the external urethral orifice. The sagittal view measured bladder neck mobility at rest and during the Valsalva maneuver at baseline and six months after the six treatments (Figure 1). Tracing the outer border of the urethral area at rest and during the Valsalva maneuver using the 3-D mode, the hypoechoic area of the proximal, middle, and distal urethras was measured (Figure 2).

Data were represented as mean ± standard deviations, and a *p* < 0.05 indicates a statistically significant difference. Statistical analyses were performed using IBM SPSS Statistical Software version 20.0 ed. Paired *t*-tests were performed for two related units on a continuous outcome. A *p*-value of less than 0.05 indicates statistical significance.

## 3. Results

### 3.1. Participants and Basic Characteristic Description

There are nineteen participants recruited in this study. The essential characteristics of patients are shown (Table 1). The mean age of participants is 55.2 ± 13.0. Their mean body mass index (BMI) is 24.0 ± 3.2. 17 out of 20 (85%) patients were in menopause. The pad test of 19 patients was 4.2 ± 5.5. After the treatment procedure of HIFEM, the pad test decreased from 4.2 ± 5.5 to 0.6 ± 1.3 (*p* = 0.045) in the 6-month follow-up.

According to their ICIQ score, 3 of them (15.8%) were divided into mild, 12 of them (63.2%) belonged to moderate, 4 of them (21.1%) were divided into severe, and 1 of them (5.3%) were divided into very severe. The definition of effectiveness represented their response as no leakage. Furthermore, 13 of them (68.4%) showed significant efficacy after HIFEM treatment in the 6-month follow-up (*p* < 0.001) (Table 2).

### 3.2. The Effect of HIFEM Treatment on Urology-Related Questionnaires

Then, some urinary-related questionnaires were inquired before the treatment of HIFEM, 1 month and 6 months after treatment, including OABSS, UDI-6, IIQ-7, ICIQ-SF, and VLQ. Our results demonstrated that the score of OABSS significantly changed from 5.3 ± 3.9 to 3.9 ± 3.6 (*p* = 0.008) and 3.6 ± 3.0 (*p* = 0.023) in the 1-month and 6-month follow-up, respectively. Additionally, the data also revealed that the score of UDI-6 significantly decreased from 35.7 ± 22.3 to 26.5 ± 20.0 (*p* = 0.013) and 15.2 ± 10.6 (*p* < 0.001) in the 1-month and 6-month follow-up, respectively.

The score IIQ-7 which represents the incontinence impact, significantly changed from 33.1 ± 28.7 to 25.5 ± 30.3 (*p* = 0.006) and 14.3 ± 17.2 (*p* = 0.005) in the 1-month and 6-month follow-up, respectively. The ICIQ-SF score also showed the same trend. It significantly changed from 9.4 ± 5.0 to 7.3 ± 4.1 (*p* = 0.008) and 5.4 ± 3.6 (*p* < 0.001) in the 1-month and 6-month follow-up, respectively. However, VLQ score did not show a significant change in both 1 and 6 months after the treatment (Table 3).

### 3.3. Urodynamics Parameters Change Under HIFEM Treatment

We then confirmed the effect of HIFEM on urodynamic change. Our data showed that Qmax (maximum flow rate) slightly increased from 20.4 ± 5.7 to 21.0 ± 8.5 without significant difference. Additionally, the residual urine (RU) did not change a lot. Moreover, first sensation to void (Vfst), maximum cystometric capacity (MCC), Pdet Qmax, detrusor pressure at peak flow (Pdet Qmax), and functional urethral length (FUL) all showed no significant difference after the HIFEM treatment in the 6-month follow-up.

However, maximum urethral closure pressure (MUCP) significantly changed from 46.4 ± 25.2 to 58.1 ± 21.2 (*p* = 0.017). Additionally, the urethral closure area (UCA) also significantly increased from 705.3 ± 302.3 to 990.0 ± 439.6 (*p* = 0.001) (Table 4).

### 3.4. Effect of HIFEM Treatment on Bladder Neck Mobility in Ultrasound Topography

We also examined bladder neck mobility and urethral area with vaginal ultrasound. Our data showed that the bladder neck did not significantly change after HIFEM treatment in the 6-month follow-up (1.2 ± 0.4 to 1.1 ± 0.3) (*p* = 0.34). Additionally, we also observed the urethral area at rest and in a strained condition. Our data showed that the treatment of HIFEM did not cause a significant change in the proximal, middle, and distal areas in the rest condition. The HIFEM treatment also did not cause a significant change in the proximal, middle, and distal areas at the maximum straining condition (Table 5).

We further measured the length and area of the vaginal and levator hiatus with ultrasound. Our data showed that only distal vaginal width significantly decreased from 4.3 ± 0.3 to 4.0 ± 0.4 (*p* = 0.05) and 4.4 ± 0.5 to 4.1 ± 0.4 (*p* = 0.04) at both rest and stress states, respectively, after the treatment of HIFEM technology. However, the vaginal area at rest state in proximal (6.1 ± 1.6 to 4.0 ± 1.2 (*p* < 0.001)), middle (5.6 ± 1.2 to 4.0 ± 1.1 (*p* = 0.002)), and distal (5.7 ± 1.4 to 4.1 ± 1.1 (*p* = 0.001)) all showed significant reduction, respectively. Moreover, the vaginal area at stress state also significantly decreased after the HIFEM treatment in proximal (6.9 ± 1.3 to 5.2 ± 2.0 (*p* = 0.002)), middle (6.5 ± 1.0 to 4.8 ± 1.6 (*p* = 0.003)), and distal (6.4 ± 1.2 to 4.4 ± 1.3 (*p* < 0.001)).

Furthermore, the levator hiatus area (12.0 ± 2.7 to 10.2 ± 2.7 (*p* = 0.028)) revealed a significant reduction at the rest state. At the stress state, the area (13.5 ± 2.8 to 11.8 ± 3.4 (*p* = 0.024)) and short axis (4.7 ± 0.5 to 4.3 ± 0.5 (*p* = 0.013)) both significantly decreased, respectively. The length of the long axis showed no difference in both rest and stress states (Table 6).

## 4. Discussion

In order to deal with SUI, lots of treatments were developed to ensure patients’ quality of life. Different treatments approached to improve the clinical symptoms of SUI from their perspectives, such as pelvic floor muscle training (PFMT), vaginal pessaries, surgical treatments (mid-urethral sling, Burch urethropexy, pubovaginal sling, artificial urethral sphincter) [31,32], or medical treatments (duloxetine) [14,33,34]. Additionally, most of these treatments focused on maintaining or strengthening the support of the pelvic floor and vaginal connective tissue. MS is one of the options. Yamanishi et al., 2019, investigated the urinary-related parameters of MS on SUI patients. Their results depicted that ICIQ-SF score, QOL score, and ALPP were significantly improved after the MS treatment in the 2.5-month follow-up [35]. Our result also revealed the effect of HIFEM on the ICIQ-SF questionnaire, and it showed a noticeable improvement in both 1-month and 6-month follow-ups (Table 3). HIFEM is a technology based on electromagnetic stimulation. Some previous research also used electrical muscle stimulation (EMS), which is usually compared to HIFEM.

EMS was first used as a therapy that could be traced back to 1745, when a German physician called Altus Kratzstein found that electrical current could control muscle, and EMS in the early stage was mostly used to treat motor paralysis patients [36]. It could stimulate involuntary contraction via different electrical current forms and frequencies [37]. Compared to HIFEM, EMS worked mainly on the surface and needed to contact the skin directly. Additionally, the FDA depicted that some unregulated EMS devices reported severe side effects such as shocks, burns, bruising, skin irritation, pain, and interference with other medical devices [38]. Additionally, due to the different principles of HIFEM and EMS, some research also compared them to the effect of muscle training [15,39,40].

Previous studies have already compared the effect of HIFEM and EMS on pelvic muscle and UI. Silantyeva et al., 2021, used 3D ultrasounds to assess anteroposterior diameter, latero-lateral diameter, and hiatal area. The result showed that only the HIFEM group showed a significant difference from the control group, and HIFEM also demonstrated a better efficacy on pelvic floor disability than the EMS group. Additionally, fewer patients were found to have urine leakage in the HIFEM group [40]. Recently more research focused on the effect of HIFEM on SUI in different aspects.

Samuels et al., 2019, demonstrated that almost 80% of their patients significantly improved from six times the treatment of HIFEM. They also evaluated their patients in ICIQ-SF score and pad test at 3-month follow-up. The ICIQ-SF score significantly decreased from 10.57 ± 4.22 to 4.16 ± 4.04 (64.42%), and the pad test significantly reduced from 2.47 ± 2.25 to 1.19 ± 1.91 (53.68%) at the 3-month follow-up. Interestingly, they recorded data after the six treatments (3 weeks). The result showed that both tests remained at a relatively low value even though no treatment was performed after 3 weeks. Additionally, a pronounced decrease in urine leakage was reported [15]. Our study also performed the ICIQ-SF and pad test at 1 month and 6 months. Our data showed that the score decreased from 9.4 ± 5.0 to 7.3 ± 4.1 (22.3%) and 5.4 ± 3.6 (42.5%) in the 1-month and 6-month follow-ups, respectively (Table 3). The pad test changed from 4.2 ± 5.5 to 0.6 ± 1.3 (85.7%) in the 6-month follow-up (Table 2). We showed a similar trend as Samuels et al., 2019, in a longer observation period [15].

Braga et al., 2022, showed that 3-tesla electromagnetic devices were treated 16 times. They found that 47% of patients significantly improved in the 2-month follow-up. The HIFEM treatment also changed the scores of the UDI-6-SF, IIQ-7, and ICIQ-SF questionnaires [41]. We also performed UDI-6, IIQ-7, and ICI-Q SF. The data in our study also showed a similar trend that ICIQ-SF showed a substantial reduction, and IIQ-7 had a smaller reduction.

Compared to Doganay et al. (2010), who evaluated magnetic stimulation effects on SUI over one year, our study uniquely integrates ultrasound-based assessments of pelvic floor structure and function. Although Doganay et al. did not report significant changes in bladder capacity or urethral closure pressure, our study observed notable improvements in MUCP and UCA, suggesting potential differences in treatment mechanisms between HIFEM and other magnetic stimulation modalities [42].

Then, Doğanay et al., 2010, demonstrated a one-year follow-up in their study. They evaluated magnetic-field stimulation in SUI and UUI. They measured pad test, voiding diary, VAS score, and urodynamic parameters. Their data showed that voiding, VAS score, and pad test all significantly reduced after the HIFEM treatment in a one-year follow-up. However, bladder capacity (Vfst), maximum cystometric capacity (MCC), and Valsalva leak point pressure (PdetQmax) did not change [42]. We also measured some urodynamic parameters in our studies, including maximum flow rate (Qmax), residual urine (RU), detrusor overactivity (DO), first sensation to void (Vfst), maximum cystometric capacity (MCC), detrusor pressure at peak flow (PdetQmax), maximum urethral closure pressure (MUCP), functional urethral length (FUL), and urethral closure area (UCA). In the 6-month follow-up, Qmax, RU, DO, Vfst, MCC, PdetQmax, and FUL did not show obvious change after the HIFEM treatment, which is similar to the study in the one-year follow-up. Only MUCP and UCA significantly increased (Table 4). HIFEM therapy provides sustained improvement in SUI symptoms, with significant effects observed up to six months post-treatment in our study. Previous literature suggested that booster sessions may help maintain therapeutic benefits over longer periods [43]. Regarding cost, HIFEM treatment is a relatively affordable, noninvasive alternative to surgical interventions, with the added benefits of minimal recovery time and low risk of complications. A detailed cost-benefit analysis comparing HIFEM to other therapies should be included in future investigations to better inform clinical and patient decisions.

Our study first used ultrasound to examine the effect of HIFEM technology on urethral topography. The data showed no significant change in all parameters, including bladder mobility, urethral area in the rest, and stress. Compared to our previous research, we suggested that the HIFEM treatment did not cause noticeable structural changes such as laser or surgery [44,45]. Additionally, the HIFEM mainly worked on the pelvic muscle instead of the urethra. Hence, we used vaginal topography to measure parameters, including vaginal width, vaginal area, and levator hiatus. To our excitement, the distal vaginal width (rest and stress), proximal vaginal area (rest and stress), middle vaginal area (rest and stress), distal vaginal area (rest and stress), levator hiatus area (rest and stress), and levator hiatus short axis (stress) all showed significant change after HIFEM treatment in the 6-month treatment (Table 6). The data revealed that HIFEM technology could significantly improve PFM integrity and enhance the function of PFM. The data on vaginal topography also echoed the result of patients’ self-assessments.

The application of HIFEM technology is considered safe and noninvasive. The device used in our study operates under restricted conditions, with patients sitting fully clothed on a chair-style applicator that delivers electromagnetic pulses. No significant adverse reactions were observed in our study cohort. However, previous studies report mild, transient discomfort during treatment as the most common side effect [15]. Due to its short developmental time, long-term safety and efficacy data remain limited.

Future studies should explore several areas to strengthen the clinical implications of HIFEM technology for treating SUI. The longer follow-up periods should be included, and the observation period should be extended to one or more years, which could provide more robust data on the long-term efficacy and sustainability of HIFEM treatment effects. Additionally, a comparative study could also be performed with other noninvasive or minimally invasive treatments, such as fractional CO_2_ laser therapy or platelet-rich plasma, which could contextualize its relative benefits and limitations. Finally, the investigation of the accurate molecular and cellular mechanisms by HIFEM on PFM might be beneficial to its application on other pelvic floor disorders. The limitations of our study must be acknowledged. The small sample size might reduce the statistical power and might not represent the whole population of SUI patients. Additionally, the lack of a control group might limit the attributed observed improvements solely to the HIFEM treatment. The short observation period was also a limitation of our study. Finally, the study was conducted in a single center, which may introduce selection bias and limit the generalizability of findings to other settings.

## 5. Conclusions

Our study showed that HIFEM treatment significantly improved SUI symptoms on pad tests and patients’ self-assessment in the 6-month follow-up. Additionally, the data from urinary-related questionnaires, including OABSS, UDI-6, IIQ-7, and ICIQ-SF, all showed a significant reduction. Then, the analysis of the urodynamic study revealed that only MUCP and UCA significantly increased after the six sessions of HIFEM treatment. The urethral and vaginal topography were performed and found that HIFEM mainly worked on PFM and enhanced its function and integrity. Our results suggest that HIFEM technology is an efficacious therapy for the treatment of SUI.

## Figures and Tables

**Figure 1 biomedicines-12-02883-f001:**
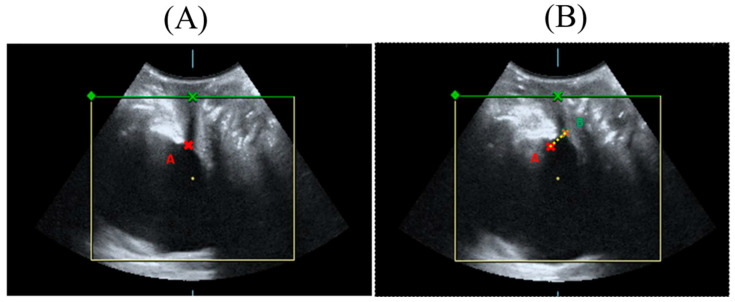
Urethral topography of the sagittal view of the bladder. Bladder neck mobility (yellow dots) from resting (**A**) to stress (**B**). Point A (red) showed the bladder neck at rest and point B (green) showed the bladder neck during straining.

**Figure 2 biomedicines-12-02883-f002:**
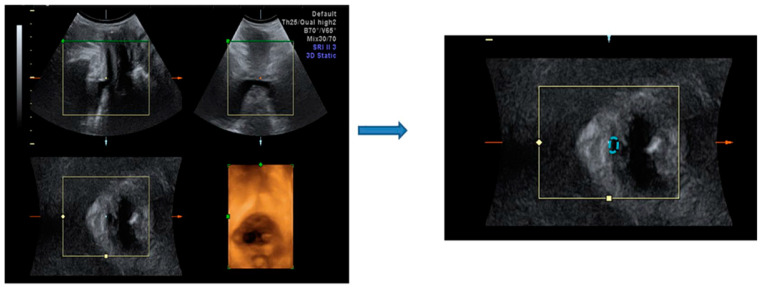
Urethral area measurement with topography. The urethral area, the hypoechoic area, is marked in the dotted line.

**Table 1 biomedicines-12-02883-t001:** Complications after SUI surgeries.

SUI Surgery Options	Complications	Reference
Pubovaginal sling	10–15% failure rate 5–10% voiding difficulties 10–15% bladder urgency or urge incontinence 5% wound or urinary tract infection 1% clotting in the legs or lungs <1% blood loss requiring transfusion <1% damage to the bladder or lower urinary tract	[10]
Burch colposuspension	<5% bladder injuries 1% urethral injury 20% voiding dysfunction 6.8~40% infection	[11]
Tension-free vaginal tape (TVT)	4.5% bladder perforation damage to pelvic blood vessels or viscera. 4% voiding difficulties and urinary retention 3–15% urgency and frequency 1.3% groin pain (1.3%)	[12]
Transobturator tape (TOT)	5–10% bladder irritability 2–3% damage to the vagina, bladder, urethra, or blood vessels 1–5% urinary tract infections 1–5% difficulty passing urine 1% risk of a wound infection 1% blood clots in the legs or chest	[13]

**Table 2 biomedicines-12-02883-t002:** Clinical background of the participants. Data are given as mean ± standard deviation or *n* (%).

*n* = 19	Pre-Tx	Post-Tx	*p* Value
Mean age (years)	55.2 ± 13.0		
Mean BMI (kg/m^2^)	24.0 ± 3.2		
Menopause	17 (89.5%)		
One hour Pad test (g)	4.2 ± 5.5	0.6 ± 1.3	0.045 *
SUI grade by ICIQ			
Mild	3 (15.8%)		
Moderate	12 (63.2%)		
Severe	4 (21.1%)		
Very severe	1 (5.3%)		
Effectiveness for SUI		13/19 (68.4%)	<0.001 *
Follow-up (months)		6M	

BMI, body mass index; SUI, stress urinary incontinence; ICIQ, International Consultation on Incontinence Questionnaire * *p* < 0.05, Student’s *t*-test.

**Table 3 biomedicines-12-02883-t003:** Questionnaire results at baseline and 6 months post-treatment.

*n* = 19	Pre-Tx	Post-Tx 1M	Post-Tx 6M	*p* Value 1M	*p* Value 3M
OABSS	5.3 ± 3.9	3.9 ± 3.6	3.6 ± 3.0	0.008 *	0.023 *
UDI-6	35.7 ± 22.3	26.5 ± 20.0	15.2 ± 10.6	0.013 *	<0.001 *
IIQ-7	33.1 ± 28.7	25.5 ± 30.3	14.3 ± 17.2	0.006 *	0.005 *
ICIQ-SF	9.4 ± 5.0	7.3 ± 4.1	5.4 ± 3.6	0.008 *	<0.001 *
VLQ	3.5 ± 0.7	3.8 ± 0.8	4.2 ± 0.8	0.363	0.189

OABSS, overactive bladder symptom score. UDI-6, urogenital distress; inventory; IIQ-7, incontinence impact questionnaire; ICIQ-SF, incontinence questionnaire-short form; VLQ, vaginal laxity questionnaire. Values are expressed as mean ± standard deviation or numbers. * Statistical significance; paired *t*-test.

**Table 4 biomedicines-12-02883-t004:** Urodynamic changes at baseline and 6 months post-treatment. Data are given as the mean ± standard deviation.

*n* = 11	Pre-Tx	Post-Tx 6M	*p* Value *
Qmax (mL/s)	20.4 ± 5.7	21.0 ± 8.5	0.968
RU (mL)	37.9 ± 44.2	28.9 ± 25.7	0.664
Vfst (mL)	142.9 ± 79.4	149.9 ± 75.2	0.913
MCC (mL)	432.2 ± 136.5	458.2 ± 112.0	0.783
Pdet Qmax (mmHg)	12.7 ± 11.9	26.7 ± 21.5	0.186
MUCP (mmHg)	46.4 ± 25.2	58.1 ± 21.2	0.017 *
FUL (cm)	28.7 ± 11.4	30.1 ± 5.1	0.795
UCA (°)	705.3 ± 302.3	990.0 ± 439.6	0.001 *

Qmax, maximum flow rate; RU, residual urine; DO, detrusor overactivity; Vfst, first sensation to void; MCC, maximum cystometric capacity; PdetQmax, detrusor pressure at peak flow; MUCP, maximum urethral closure pressure; FUL, functional urethral length; UCA, urethral closure area. * Statistical significance; paired *t*-test.

**Table 5 biomedicines-12-02883-t005:** Urethral topography parameters at baseline and 6 months after treatment during resting and straining.

*n* = 14		Pre-Tx	Post-Tx 6M	*p* Value
Bladder neck mobility (mm)		1.2 ± 0.4	1.1 ± 0.3	0.34
Urethral area (mm²)	proximal	0.8 ± 0.4	0.7 ± 0.2	0.58
Resting	middle	0.8 ± 0.3	0.8 ± 0.2	0.17
	distal	2.3 ± 1.0	2.5 ± 1.2	0.44
Urethral area (mm²)	proximal	4.1 ± 1.3	5.3 ± 17.2	0.34
straining	middle	0.8 ± 0.2	0.8 ± 0.3	0.17
	distal	0.7 ± 0.2	0.6 ± 0.2	0.19

**Table 6 biomedicines-12-02883-t006:** Changes in vaginal topography before and 6 months after HIFEM technology treatment.

*n* = 14		Rest			Stress		
		Pre-Tx	Post-Tx 6M	*p* Value	Pre-Tx	Post-Tx 6M	*p* Value
Vaginal width	Proximal	4.6 ± 0.5	4.4 ± 0.5	0.75	4.6 ± 0.5	4.4 ± 0.6	0.67
	Middle	4.2 ± 0.4	4.2 ± 0.5	0.44	4.4 ± 0.5	4.2 ± 0.5	0.19
	Distal	4.3 ± 0.3	4.0 ± 0.4	0.05 *	4.4 ± 0.5	4.1 ± 0.4	0.04 *
Vaginal area	Proximal	6.1 ± 1.6	4.0 ± 1.2	<0.001 *	6.9 ± 1.3	5.2 ± 2.0	0.002 *
	Middle	5.6 ± 1.2	4.0 ± 1.1	0.002 *	6.5 ± 1.0	4.8 ± 1.6	0.003 *
	Distal	5.7 ± 1.4	4.1 ± 1.1	0.001 *	6.4 ± 1.2	4.4 ± 1.3	<0.001 *
Levator hiatus	Area	12.0 ± 2.7	10.2 ± 2.7	0.028 *	13.5 ± 2.8	11.8 ± 3.4	0.024 *
	Short axis	4.5 ± 0.5	4.2 ± 0.5	0.104	4.7 ± 0.5	4.3 ± 0.5	0.013 *
	Long axis	3.5 ± 0.6	3.4 ± 0.6	1.0	3.9 ± 0.6	3.7 ± 0.6	0.5

* Statistical significance; paired *t*-test.

## Data Availability

The data presented in this study are available upon request from the corresponding authors. The data are not publicly available due to privacy.

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
