# Peer review of "Effect of High-Intensity Focused Electromagnetic Technology in the Treatment of Female Stress Urinary Incontinence"

_biomedicines, 2024, doi:10.3390/biomedicines12122883_

Round 1
Reviewer 1 Report
Comments and Suggestions for Authors
All abbreviations used in the Abstract need to be spelled out.
Please provide the rationale for the frequency and duration of the intervention. More details regarding the HIFEM parameters used in this study need to be added.
Twenty female participants were recruited in this study, however, there is no justification for the sample size determination. Why was this sample size considered adequate for this study?
Lines 235-239: Why was the paper by Alazab et al. 2022 described here? It focuses on a wholly different condition and population (stroke-induced shoulder subluxation) which is not comparable to the present study.
Please follow the TIDieR (Template for Intervention Description and Replication) checklist to improve the quality of reporting and the replicability of interventions. A completed checklist should be included in the submission files.
Significant problem- the Materials and Methods session is incomplete, there is no information regarding statistical analysis.
Detailed recommendations for future studies are needed to strengthen the clinical implications of this study. Limitations of this study should be recognized.
The last paragraph of the Discussion is duplicated in content with the Conclusion.
The last section of this manuscript "6. Patents" appears irrelevant and should be removed.
Comments on the Quality of English LanguageAn English-proof read of the entire manuscript is recommended since numerous syntax errors were detected, compromising the readability.
Author Response
Q1. All abbreviations used in the Abstract need to be spelled out.
Our reply: Thanks for the careful review, we corrected the abstract as followed “Background: The aim of the study was to assess the effect of High-Intensity Focused Electro-magnetic (HIFEM) technology in the treatment of female stress urinary incontinence (SUI). Materials and methods: 20 women with SUI were delivered a treatment course with HIFEM technology. Patients attended 6 therapies scheduled twice a week. Validated questionnaires were assessed, including overactive bladder symptoms score (OABSS), urogenital distress inventory-6 (UDI-6), incontinence impact questionnaire-7 (IIQ-7), international consultation on incontinence questionnaire (ICIQ), and valued living questionnaire (VLQ). Some urodynamic parameters such as maximum flow rate (Qmax), residual urine (RU), bladder volume at first sensation to void (Vfst). Bladder neck mobility in ultrasound topography was also collected pre-, post-treatment, at 1- and 6-month follow-up visits. Results: HIFEM treatment significantly improved SUI symptoms on pad tests from 4.2±5.5 to 0.6±1.3 and patients’ self-assessment in the 6-month follow-up. Besides, the data from urinary-related questionnaires, including OABSS (5.3±3.9 to 3.9±3.6), UDI-6 (35.7±22.3 to 15.2±10.6), IIQ-7 (33.1±28.7 to 14.3±17.2), and ICIQ (9.4±5.0 to 5.4±3.6), all showed a significant reduction. Then, the analysis of the urodynamic study revealed that only maximum urethral closure pressure (MUCP) (46.4±25.2 to 58.1±21.2) and urethral closure angle (UCA) (705.3±302.3 to 990.0±439.6) significantly increased after the six sessions of HIFEM treatment. The urethral and vaginal topography were performed, and found that HIFEM mainly worked on pelvic floor muscles (PFM) and enhanced its function and integrity. Conclusion: The results suggest that HIFEM technology is an efficacious therapy for treatment of SUI”
Q2. Please provide the rationale for the frequency and duration of the intervention. More details regarding the HIFEM parameters used in this study need to be added.
Our reply: Thanks for the careful review, we added the paragraph as followed in material and methods: “The HIFEM (BTL EMSELLA® non-invasive therapeutic device) treatment is performed twice a week and sustained for three weeks. Each treatment lasts for 28 minutes. The device uses a chair-style applicator in which the fully clothed patients can directly sit on the center of the stimulation coil, which generates a 2.5 Tesla magnetic field. The rationale behind the frequency choice in HIFEM technology is to optimize the induction of supramaximal muscle contractions and fat reduction. The frequency is selected for ensure effective penetration and interaction with PFM. This is achieved by using low-frequency electromagnetic waves that penetrate deep into the muscle tissue [1]. The frequency is calibrated to stimulate muscle fibers effectively, promoting increased muscle mass and strength, which is beneficial for both aesthetic and functional improvements [2].”
- Mezzana, P.; Pieri, L.; Leone, A.; Fusco, I. Schwarzy: The new system for muscle toning and body shaping. Journal of Cosmetic Dermatology 2021, 20, 2678-2680, doi:https://doi.org/10.1111/jocd.14275.
- Goldberg, D.J.; Enright, K.M.; Goldfarb, R.; Katz, B.; Gold, M. The role and clinical benefits of high-intensity focused electromagnetic devices for non-invasive lipolysis and beyond: A narrative review and position paper. J Cosmet Dermatol 2021, 20, 2096-2101, doi:10.1111/jocd.14203.
Q3. Twenty female participants were recruited in this study, however, there is no justification for the sample size determination. Why was this sample size considered adequate for this study?
Our reply: Thank you for your valuable comment, we acknowledge the importance of ensuring that the sample size is adequate to provide reliable and interpretable results. Our study was designed as a pilot study to explore effect of HIFEM in the treatment of female stress urinary incontinence. Pilot studies often involve smaller sample sizes as their primary objective is to assess feasibility, optimize methodologies, and gather initial data for future larger-scale studies. Besides, some studies also recruited about twenty patients in their SUI related literature [1,2]. Besides, the collected data provided sufficient statistical power to detect clinically meaningful changes or trends in SUI. Additionally, the findings from this study will serve as a foundation for future research, where a larger sample size will be calculated based on the effect sizes observed in this study.
- Goldstein, S.; Williams, J.; Kim, N.; Goldstein, I. 049 A Double-Blind, Sham Controlled Prospective Pilot Study of Urinary Stress Incontinence and Sexual Function in Women After 6 Treatments with HIFEM Technology (Emsella): Interim Analysis. The Journal of Sexual Medicine 2020, 17, S242, doi:https://doi.org/10.1016/j.jsxm.2020.04.285.
- Ong, H.L.; Sokolova, I.; Bekarma, H.; Curtis, C.; Macdonald, A.; Agur, W. Development, validation and initial evaluation of patient-decision aid (SUI-PDA©) for women considering stress urinary incontinence surgery. Int Urogynecol J 2019, 30, 2013-2022, doi:10.1007/s00192-019-04047-z.
We agree that future studies should incorporate more patients based on the effect sizes observed here to further validate our findings.
Q4. Lines 235-239: Why was the paper by Alazab et al. 2022 described here? It focuses on a wholly different condition and population (stroke-induced shoulder subluxation) which is not comparable to the present study.
Our reply: Thanks for the careful review, we deleted the paragraph as your suggestion.
Q5. Please follow the TIDieR (Template for Intervention Description and Replication) checklist to improve the quality of reporting and the replicability of interventions. A completed checklist should be included in the submission files.
Our reply: Thanks for the valuable suggestion, we added a TIDieR checklist in supplemental file as your suggestions.
Q6. Significant problem- the Materials and Methods session is incomplete, there is no information regarding statistical analysis.
Our reply: Thanks for the careful review, we added the statistical analysis section as follows “Data were represented as mean ± standard deviations, and a p < 0.05 indicates a statistically significant difference. Statistical analyses were performed using IBM SPSS Statistical Software version 20.0 ed. Paired t-tests were performed for two related units on a continuous outcome. A p-value of less than 0.05 indicates statistical significance.”
Q7. Detailed recommendations for future studies are needed to strengthen the clinical implications of this study. Limitations of this study should be recognized.
Our reply: Thanks for the valuable suggestion, we added the paragraph as follows “The future studies should explore several areas to strengthen the clinical implications of HIFEM technology for treating SUI. The longer follow-up periods should be included and extended the observation period to one or more years that could provide more robust data on the long-term efficacy and sustainability of HIFEM treatment effects. Besides, a comparative study could also be performed with other non-invasive or minimally invasive treatments, such as fractional CO2 laser therapy or platelet-rich plasma, could contextualize its relative benefits and limitations. Finally, the investigation of the accurate molecular and cellular mechanisms by HIFEM on PFM might be beneficial to its application on other pelvic floor disorders. The limitations of our study must be acknowledged. The small sample size might reduce the statistical power and might not represent the whole population of SUI patients. Besides, the lack of a control group might limit the attribute observed improvements solely to the HIFEM treatment. The short observation period was also a limitation of our study. Finally, the study was conducted in a single center which may introduce selection bias and limit the generalizability of findings to other settings.”
Q8. The last paragraph of the Discussion is duplicated in content with the Conclusion.
Our reply: Thanks for the careful remind, we deleted the paragraph.
The last section of this manuscript "6. Patents" appears irrelevant and should be removed.
Our reply: Thanks for the careful remind, we deleted the section as your suggestion.
Reviewer 2 Report
Comments and Suggestions for Authors
1. Full name of all abbreviation in Abstract should be stated.
2. Numerical value of main findings should be revealed in Abstract.
3. Please indicate in Introduction, why your research work is in the scope of Biomedicines.
4. Please add punctuation before and after +- in all Tables and in content.
5. Please modify your footnote of all data as usual international format.
6. The suggestion and limitation of this research should be included in Conclusion.
7. The explanation more on principle of HIFEM with its function and mechanism of action should be included in Introduction. The safety and control condition of applying this technique should be stated in the content with how about the long-term use affect and side effect and adverse reaction. How long action could be achieved from the treatment and cost of treatment should be recommended and compared or suggested.
8. More detail for HIFEM (BTL EMSELLA® non-invasive therapeutic device) should be revealed such as brand, model, company, city, state, country.
9. Ethical approval pass should be informed.
10. The more discussion with comparative with relative or previous published article should be addressed.
Author Response
Reviewer 2.
- Full name of all abbreviation in Abstract should be stated.
Our reply: Thanks for the careful review, we corrected the abstract as follows “Background: The aim of the study was to assess the effect of High-Intensity Focused Electro-magnetic (HIFEM) technology in the treatment of female stress urinary incontinence (SUI). Materials and methods: 20 women with SUI were delivered a treatment course with HIFEM technology. Patients attended 6 therapies scheduled twice a week. Validated questionnaires were assessed, including overactive bladder symptoms score (OABSS), urogenital distress inventory-6 (UDI-6), incontinence impact questionnaire-7 (IIQ-7), international consultation on incontinence questionnaire (ICIQ), and valued living questionnaire (VLQ). Some urodynamic parameters such as maximum flow rate (Qmax), residual urine (RU), bladder volume at first sensation to void (Vfst). Bladder neck mobility in ultrasound topography was also collected pre-, post-treatment, at 1- and 6-month follow-up visits. Results: HIFEM treatment significantly improved SUI symptoms on pad tests from 4.2±5.5 to 0.6±1.3 and patients’ self-assessment in the 6-month follow-up. Besides, the data from urinary-related questionnaires, including OABSS (5.3±3.9 to 3.9±3.6), UDI-6 (35.7±22.3 to 15.2±10.6), IIQ-7 (33.1±28.7 to 14.3±17.2), and ICIQ (9.4±5.0 to 5.4±3.6), all showed a significant reduction. Then, the analysis of the urodynamic study revealed that only maximum urethral closure pressure (MUCP) (46.4±25.2 to 58.1±21.2) and urethral closure angle (UCA) (705.3±302.3 to 990.0±439.6) significantly increased after the six sessions of HIFEM treatment. The urethral and vaginal topography were performed, and found that HIFEM mainly worked on pelvic floor muscles (PFM) and enhanced its function and integrity. Conclusion: The results suggest that HIFEM technology is an efficacious therapy for treatment of SUI”
- Numerical value of main findings should be revealed in Abstract.
Our reply: Thanks for the valuable suggestion, we corrected the abstract as follows “Background: The aim of the study was to assess the effect of High-Intensity Focused Electro-magnetic (HIFEM) technology in the treatment of female stress urinary incontinence (SUI). Materials and methods: 20 women with SUI were delivered a treatment course with HIFEM technology. Patients attended 6 therapies scheduled twice a week. Validated questionnaires were assessed, including overactive bladder symptoms score (OABSS), urogenital distress inventory-6 (UDI-6), incontinence impact questionnaire-7 (IIQ-7), international consultation on incontinence questionnaire (ICIQ), and valued living questionnaire (VLQ). Some urodynamic parameters such as maximum flow rate (Qmax), residual urine (RU), bladder volume at first sensation to void (Vfst). Bladder neck mobility in ultrasound topography was also collected pre-, post-treatment, at 1- and 6-month follow-up visits. Results: HIFEM treatment significantly improved SUI symptoms on pad tests from 4.2±5.5 to 0.6±1.3 and patients’ self-assessment in the 6-month follow-up. Besides, the data from urinary-related questionnaires, including OABSS (5.3±3.9 to 3.9±3.6), UDI-6 (35.7±22.3 to 15.2±10.6), IIQ-7 (33.1±28.7 to 14.3±17.2), and ICIQ (9.4±5.0 to 5.4±3.6), all showed a significant reduction. Then, the analysis of the urodynamic study revealed that only maximum urethral closure pressure (MUCP) (46.4±25.2 to 58.1±21.2) and urethral closure angle (UCA) (705.3±302.3 to 990.0±439.6) significantly increased after the six sessions of HIFEM treatment. The urethral and vaginal topography were performed, and found that HIFEM mainly worked on pelvic floor muscles (PFM) and enhanced its function and integrity. Conclusion: The results suggest that HIFEM technology is an efficacious therapy for treatment of SUI”
- Please indicate in Introduction, why your research work is in the scope of Biomedicines.
Our reply: Thank you for the insightful comment. We clarified how our study aligns with the scope of Biomedicines and revised the Introduction section to explicitly address this point. We have emphasized some aspects such as translational research, which investigated the clinical application of HIFEM technology as a non-invasive therapeutic modality for SUI. It contributed to the development of innovative therapeutic strategies, which aligns with the journal’s focus on translational medical research and clinical applications. Besides, the evaluation of the effects of HIFEM on pelvic floor muscle function, urodynamic parameters, and imaging-based assessments, we address key areas such as pathogenesis mechanisms and targeted therapy, which are central to Biomedicines. We have incorporated this section into the revised Introduction to make the alignment with Biomedicines more explicit as follows. Thank you for pointing out this opportunity to enhance the manuscript.
“Our study contributes to translational medical research by evaluating a novel therapeutic strategy aimed at addressing the pathophysiology of SUI. Besides, it provides mechanistic insights into the role of neuromodulation in pelvic floor muscle rehabilitation, supported by urodynamic and imaging-based analyses. Finally, the findings underscore the potential of HIFEM technology in regenerative medicine, offering a non-invasive alternative to existing treatments for SUI.”
- Please add punctuation before and after +- in all Tables and in content.
Our reply: Thanks for the careful review, we have added the punctuation as your suggestion in all Tables and in content.
- Please modify your footnote of all data as usual international format.
Our reply: Thanks for the careful review, we adjusted the usual international format of our manuscript as your suggestion.
- The suggestion and limitation of this research should be included in Conclusion.
Our reply: The suggestion and limitation of our research were replenished as follows: “The future studies should explore several areas to strengthen the clinical implications of HIFEM technology for treating SUI. The longer follow-up periods should be included and extended the observation period to one or more years that could provide more robust data on the long-term efficacy and sustainability of HIFEM treatment effects. Besides, a comparative study could also be performed with other non-invasive or minimally invasive treatments, such as fractional CO2 laser therapy or platelet-rich plasma, could contextualize its relative benefits and limitations. Finally, the investigation of the accurate molecular and cellular mechanisms by HIFEM on PFM might be beneficial to its application on other pelvic floor disorders. The limitations of our study must be acknowledged. The small sample size might reduce the statistical power and might not represent the whole population of SUI patients. Besides, the lack of a control group might limit the attribute observed improvements solely to the HIFEM treatment. The short observation period was also a limitation of our study. Finally, the study was conducted in a single center which may introduce selection bias and limit the generalizability of findings to other settings.”
- The explanation more on principle of HIFEM with its function and mechanism of action should be included in Introduction. The safety and control condition of applying this technique should be stated in the content with how about the long-term use affect and side effect and adverse reaction. How long action could be achieved from the treatment and cost of treatment should be recommended and compared or suggested.
Our reply: Thank you for the valuable comments. We added the following paragraph to further state the principle and mechanism of action of HIFEM as followed : “These contractions surpass those achievable through voluntary muscle activation, promoting muscle strengthening, enhanced neuromuscular control, and restoration of pelvic floor function. The mechanism involves the depolarization of motor neurons, leading to repeated and controlled muscle contractions, which improve the structural support of the urethra and reduce urinary incontinence symptoms.”
The safety and long-term effects were added as followed in discussion section “The application of HIFEM technology is considered safe and non-invasive. The device used in our study operates under restricted conditions, with patients sitting fully clothed on a chair-style applicator that delivers electromagnetic pulses. No significant adverse reactions were observed in our study cohort. However, previous studies report mild, transient discomfort during treatment as the most common side effect [1]. Due to its short developmental time, long-term safety and efficacy data remain limited.”
The duration of treatment effects was also added in the discussion section as follows: “HIFEM therapy provides sustained improvement in SUI symptoms, with significant effects observed up to six months post-treatment in our study. Previous literature suggested that booster sessions may help maintain therapeutic benefits over longer periods [2]. Regarding cost, HIFEM treatment is a relatively affordable, non-invasive alternative to surgical interventions, with the added benefits of minimal recovery time and low risk of complications. A detailed cost-benefit analysis comparing HIFEM to other therapies should be included in future investigations to better inform clinical and patient decisions.”
- Samuels, J.B.; Pezzella, A.; Berenholz, J.; Alinsod, R. Safety and Efficacy of a Non-Invasive High-Intensity Focused Electromagnetic Field (HIFEM) Device for Treatment of Urinary Incontinence and Enhancement of Quality of Life.
- Brandeis, J. (126) Application of HIFEM Therapy Improves Orgasmic, Urinary and Erectile Function by Stimulation of Pelvic Floor Muscles. The Journal of Sexual Medicine 2024, 21, qdae001.120, doi:10.1093/jsxmed/qdae001.120.
- More detail for HIFEM (BTL EMSELLA® non-invasive therapeutic device) should be revealed such as brand, model, company, city, state, country.
Our reply: Thanks for the careful review, we added the following paragraph in the material and methods section
“We performed HIFEM treatment with BTL EMSELLA (BTL Industries Inc, Boston, MA) device. BTL EMSELLA uses HIFEM technology for PFM strengthening and reduction of SUI. The device is consisted of a power generator and a circular coil mounted in the seat of the chair.”
- Ethical approval pass should be informed.
Our reply: Thanks for the valuable comments, the ethical approval pass were listed in the following paragraph
“Institutional Review Board Statement: The study was conducted in accordance with the Decla-ration of Helsinki, and approved by the Institutional Review Board of Kaohsiung Medical University Chung-Ho Memorial Hospital (KMUHIRB-F(I)-20210133).
Informed Consent Statement: Informed consent was obtained from all subjects involved in the study.”
- The more discussion with comparative with relative or previous published article should be addressed.
Our reply: Thanks for the valuable comments, we have revised the Discussion section to include a more comprehensive comparison of our findings with previous studies and related literature as followed
“Compared to Doganay et al. (2010), who evaluated magnetic stimulation effects on SUI over one year, our study uniquely integrates ultrasound-based assessments of pelvic floor structure and function. Although Doganay et al. did not report significant changes in bladder capacity or urethral closure pressure, our study observed notable improvements in MUCP and UCA, suggesting potential differences in treatment mechanisms between HIFEM and other magnetic stimulation modalities [1].”
- Doğanay, M.; Kilic, S.; Yilmaz, N. Long-term effects of extracorporeal magnetic innervations in the treatment of women with urinary incontinence: results of 3-year follow-up. Archives of Gynecology and Obstetrics 2010, 282, 49-53, doi:10.1007/s00404-009-1243-5.
Round 2
Reviewer 1 Report
Comments and Suggestions for Authors
I am satisfied that you have adequately addressed the requested revisions. Thank you.